# Does mitigating ML's impact disparity require treatment disparity?

**Zachary C. Lipton**[1]**, Alexandra Chouldechova**[1]**, Julian McAuley**[2]
[1]Carnegie Mellon University
[2]University of California, San Diego
zlipton@cmu.edu, achould@cmu.edu, jmcauley@cs.ucsd.edu

## Abstract

Following precedent in employment discrimination law, two notions of disparity are widely-discussed in papers on fairness and ML. Algorithms exhibit *treatment disparity* if they formally treat members of protected subgroups differently; algorithms exhibit *impact disparity* when outcomes differ across subgroups (even unintentionally). Naturally, we can achieve impact parity through purposeful treatment disparity. One line of papers aims to reconcile the two parities proposing *disparate learning processes* (DLPs). Here, the sensitive feature is used during training but a *group-blind* classifier is produced. In this paper, we show that: (i) when sensitive and (nominally) nonsensitive features are correlated, DLPs will indirectly implement treatment disparity, undermining the policy desiderata they are designed to address; (ii) when group membership is *partly* revealed by other features, DLPs induce within-class discrimination; and (iii) in general, DLPs provide suboptimal trade-offs between accuracy and impact parity. Experimental results on several real-world datasets highlight the practical consequences of applying DLPs.

## 1 Introduction

Effective decision-making requires choosing among options given the available information. That much is unavoidable, unless we wish to make trivial decisions. In selection processes, such as hiring, university admissions, and loan approval, the options are people; the available features include (but are rarely limited to) direct evidence of qualifications; and decisions impact lives.

Laws in many countries restrict the ways in which certain decisions can be made. For example, Title VII of the US Civil Rights Act [1], forbids employment decisions that discriminate on the basis of certain *protected characteristics*. Interpretation of this law has led to two notions of discrimination: *disparate treatment* and *disparate impact*. *Disparate treatment* addresses intentional discrimination, including (i) decisions explicitly based on protected characteristics; and (ii) intentional discrimination via proxy variables (e.g literacy tests for voting eligibility). *Disparate impact* addresses facially neutral practices that might nevertheless have an "unjustified adverse impact on members of a protected class" [1]. One might hope that detecting unjustified impact were as simple as detecting unequal outcomes. However, absent intentional discrimination, unequal outcomes can emerge due to correlations between protected and unprotected characteristics. Complicating matters, unequal outcomes may not always signal unlawful discrimination [2].

Recently, owing to the increased use of machine learning (ML) to assist in consequential decisions, the topic of quantifying and mitigating ML-based discrimination has attracted interest in both policy and ML. However, while the existing legal doctrine offers qualitative ideas, intervention in an ML-based system requires more concrete formalism. Inspired by the relevant legal concepts, technical papers have proposed several criteria to quantify discrimination. One criterion requires that the fraction given a positive decision be equal across different groups. Another criterion states that a

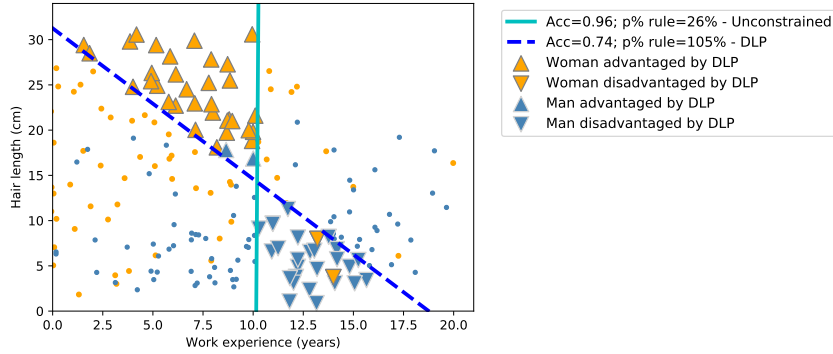

Figure 1: Demonstration of a DLP's undesirable side effects on a simple example of hiring data (see §4.1). An unconstrained classifier (vertical line) hires candidates based on work experience, yielding higher hiring rates for men than for women. A DLP (dashed diagonal) achieves near-parity by differentiating based on an irrelevant attribute (hair length). The DLP *hurts* some short-haired women, flipping their decisions to reject, and helps some long-haired men.

classifier should be blind to the protected characteristic. Within the technical literature, these criteria are commonly referred to as *disparate impact* and *disparate treatment*, respectively.

In this paper, we call these technical criteria *impact parity* and *treatment parity* to distinguish them from their legal antecedents. The distinction between technical and legal terminology is important to maintain. While impact and treatment parity are inspired by legal concepts, technical approaches that achieve these criteria may fail to satisfy the underlying legal and ethical desiderata.

We demonstrate one such disconnect through DLPs, a class of algorithms designed to simultaneously satisfy treatment- and impact-parity criteria [3–5]. DLPs operate according to the following principle: *The protected characteristic may be used during training, but is not available to the model at prediction time.* In the earliest such approach the protected characteristic is used to winnow the set of acceptable rules from an expert system [3]. Others incorporate the protected characteristic as either a regularizer, a constraint, or to preprocess the training data [5–7].

These approaches are grounded in the premise that DLPs are acceptable in cases where using a protected characteristic as a direct input to the model would constitute *disparate treatment* and thus be impermissible. Indeed, DLPs in some sense operationalize a form of prospective fair "test design" that is well aligned with the ruling in *Ricci v. DeStefano* [8]. In this paper we investigate the utility of DLPs as a technical solution and present the following cautionary insights:

1. When protected characteristics are redundantly encoded in the other features, sufficiently powerful DLPs can (indirectly) implement any form of treatment disparity.
2. When protected characteristics are partially encoded DLPs induce within-class discrimination based on irrelevant features, and can harm some members of the protected group.
3. DLPs provide a suboptimal trade-off between accuracy and impact parity.
4. While disparate treatment is by definition illegal, the status of treatment disparity is debated [9].

## 2  Disparate Learning Processes

To begin our formal description of the prior work, we'll introduce some notation. A dataset consists of $n$ *examples*, or *data points* $\{\mathbf{x}_i \in \mathcal{X}, y_i \in \mathcal{Y}\}$, each consisting of a feature vector $\mathbf{x}_i$ and a label $y_i$. A supervised learning algorithm $f : \mathcal{X}^n \times \mathcal{Y}^n \to (\mathcal{X} \to [0,1])$ is a mapping from datasets to models. The learning algorithm produces a model $\hat{y} : \mathcal{X} \to \mathcal{Y}$, which given a feature vector $\mathbf{x}_i$, predicts the corresponding output $y_i$. In this discussion we focus on binary classification ($\mathcal{Y} = \{0,1\}$).

We consider probabilistic classifiers which produce estimates $\hat{p}(\mathbf{x})$ of the conditional probability $\mathbb{P}(y = 1 \mid \mathbf{x})$ of the label given a feature vector $\mathbf{x}$. To make a prediction $\hat{y}(\mathbf{x}) \in \mathcal{Y}$ given an estimated probability $\hat{p}(\mathbf{x})$ a threshold rule is used: $\hat{y}_i = 1$ iff $\hat{p}_i > t$. The optimal choice of the threshold $t$ depends on the performance metric being optimized. In our theoretical analysis, we consider

optimizing the *immediate utility* [10], of which classification accuracy (expected $0 - 1$ loss) is a special case. We will define this metric more precisely in the next section.

In formal descriptions of discrimination-aware ML, a dataset possesses a protected feature $z_i \in \mathcal{Z}$, making each example a three-tuple $(\mathbf{x}_i, y_i, z_i)$. The protected characteristic may be real-valued, like age, or categorical, like race or gender. The goal of many methods in discrimination-aware ML is not only to maximize accuracy, but also to ensure some form of impact parity. Following related work, we consider binary protected features that divide the set of examples into two groups $a$ and $b$. Our analysis extends directly to settings with more than two groups.

Of the various measures of impact disparity, the two that are the most relevant here are the Calders-Verwer gap and the p-% rule. At a given threshold $t$, let $q_z = \frac{1}{n_z} \sum_{i:z_i=z} \mathbb{1}(\hat{p}_i > t)$, where $n_z = \sum_i^n \mathbb{1}(z_i = z)$. The **Calders-Verwer (CV) gap**, $q_a - q_b$, is the difference between the proportions assigned to the positive class in the advantaged group $a$ and the disadvantaged group $b$ [4]. The p-% rule is a related metric[5]. Classifiers satisfy the **p-% rule** if $q_b/q_a \geq p/100$.

Many papers in discrimination-aware ML propose to optimize accuracy (or some other risk) subject to constraints on the resulting level of impact parity as assessed by some metric [3, 7, 11–14]. Use of DLPs presupposes that using the protected feature $z$ as a model input is impermissible in this effort. Discarding protected features, however, does not guarantee impact parity [15]. DLPs incorporate $z$ in the learning algorithm, but without making it an input to the classifier. Formally, a DLP is a mapping: $\mathcal{X}^n \times \mathcal{Y}^n \times \mathcal{Z}^n \to (\mathcal{X} \to \mathcal{Y})$. By definition, DLPs achieve treatment parity. However, satisfying *treatment parity* in this fashion may still violate *disparate treatment*.

**Alternative approaches.** Researchers have proposed a number of other techniques for reconciling accuracy and impact parity. One approach consists of preprocessing the training data to reduce the dependence between the resulting model predictions and the sensitive attribute [6, 16–19]. These methods differ in terms of which variables they affect and the degree of independence achieved. [6] proposed flipping negative labels of training examples form the protected group. [20] proposed learning representations (cluster assignments) so that group membership cannot be inferred from cluster membership. [17] and [19] also construct representations designed to be marginally independent from $Z$.

## 3   Theoretical Analysis

We present a set of simple theoretical results that demonstrate the optimality of treatment disparity, and highlight some properties of DLPs. We summarize our results as follows:

1. Direct treatment disparity on the basis of $z$ is the optimal strategy for maximizing classification accuracy[1] subject to CV and $p$-% constraints.
2. When $X$ fully encodes $Z$, a sufficiently powerful DLP is equivalent to treatment disparity.

In Section 4, we empirically demonstrate a related point:

3. When $X$ only partially encodes $Z$, a DLP may be suboptimal and can induce intra-group disparity on the basis of otherwise irrelevant features correlated with $Z$.

**Treatment disparity is optimal** Absent impact parity constraints, the Bayes-optimal decision rule for minimizing expected $0 - 1$ loss (i.e., maximizing accuracy) is given by $d^*_{\mathrm{uncon}}(\mathbf{x}, z) = \delta(p_{Y|X,Z}(\mathbf{x}, z) \geq 0.5)$, where $\delta()$ is an indicator function.

We now show that the optimal decision rules in the CV and $p$-% constrained problems have a similar form. The optimal decision rule will again be based on thresholding $p_{Y|X,Z}(\mathbf{x}, z)$, but at *group-specific thresholds*. These rules can be thought of as operationalizing the following mechanism: Suppose that we start with the classifications of the unconstrained rule $d^*_{\mathrm{uncon}}(\mathbf{x}, z)$, and this results in a CV gap of $q_a - q_b > \gamma$. To reduce the CV gap to $\gamma$ we have two mechanisms: We can (i) flip predictions from 0 to 1 in group $b$, and (ii) we can flip predictions from 1 to 0 in group $a$. The optimal strategy is to perform these flips on group $b$ cases that have the highest value of $p_{Y|X,Z}(\mathbf{x}, z)$ and group $a$ cases that have the lowest value of $p_{Y|X,Z}(\mathbf{x}, z)$.

The results in this section adapt the work of [10], who establish optimal decision rules $d$ under exact parity. In that work, the authors characterize the optimal decision rule $d = d(\mathbf{x}, z)$ that maximizes the *immediate utility* $u(d, c) = \mathbb{E}[Yd(X, Z) - cd(X, Z)]$ for $(0 < c < 1)$, under different exact parity criteria. We begin with a lemma showing that expected classification accuracy has the functional form of an immediate utility function.

**Lemma 1.** *Optimizing classification accuracy is equivalent to optimizing immediate utility with $c = 0.5$.*

*Proof.* The expected accuracy of a binary decision rule $d(X)$ can be written as $\mathbb{E}[Yd(X) + (1 - Y)(1 - d(X))]$. Expanding and rearranging this expression gives

$$\mathbb{E}[Yd(X) + (1 - Y)(1 - d(X))] = \mathbb{E}(2Yd(X) - d(X)) + \mathbb{E}(Y) + 1 = 2u(d, 0.5) + \mathbb{E}(Y) + 1.$$

The only term in this expression that depends on $d$ is the immediate utility $u$. Thus the decision rule that maximizes $u$ also maximizes accuracy. $\qquad\square$

We note that the results in this section are related to the recent independent work of [21], who derive Bayes-optimal decision rules under the same parity constraints we consider here, working instead with the *cost-sensitive risk*, $\mathrm{CS}(d; c) = \pi(1 - c)\mathrm{FNR}(d) + (1 - \pi)c\mathrm{FPR}(d)$, where $\pi = \mathbb{P}(Y = 1)$. One can show that $u(d, c) = -\mathrm{CS}(d; c) + \pi(1 - c)$, and hence the problem of maximizing immediate utility considered here is equivalent to minimizing cost-sensitive risk as in [21]. In our case, it will be more convenient to work with the immediate utility.

For the next set of results, we follow [10] and assume that $p_{Y|X,Z}(X, Z)$, viewed as a random variable, has positive density on $[0, 1]$. This ensures that the optimal rules are unique and deterministic by disallowing point-masses of probability that would necessitate tie-breaking among observations with equal probability. The first result that we state is a direct corollary of two results in [10]. It considers the case where we desire exact parity, i.e., that $q_a = q_b$.

**Corollary 2.** *The optimal decision rules $d^*$ under various parity constraints have the following form and are unique up to a set of probability zero:*

1. *Among rules satisfying statistical parity (the 100% rule), the optimum is $d^*(\mathbf{x}, z) = \delta(p_{Y|X,Z}(\mathbf{x}, z) \geq t_z)$, where $t_z \in [0, 1]$ are constants that depend only on group membership $z$.*
2. *Among rules that have equal false positive rates across groups, the optimum is $d^*(\mathbf{x}, z) = \delta(p_{Y|X,Z}(\mathbf{x}, z) \geq s_z)$, where $s_z$ are constants that depend only on group membership $z$ (but are different from $t_z$).*
3. *(1) and (2) continue to hold even in the resource-constrained setting where the overall proportion of cases classified as positive is constrained.*

*Proof.* (1) and (2) are direct corollaries of Lemma 1 combined with Thm 3.2 and Prop 3.3 of [10]. $\qquad\square$

The next set of results establishes optimality under general $p$-% and CV rules.

**Proposition 3.** *Under the same assumptions as above, the optimum among rules that satisfy the CV constraint $0 \leq q_a - q_b < \gamma$ or the $p$-% rule also has the form $d^*(\mathbf{x}, z) = \delta(p_{Y|X,Z}(\mathbf{x}, z) \geq t_z)$, where $t_z \in [0, 1]$ are constants that depend on the group membership $z$, and on the choice of constraint parameter $\gamma$ or $p$. The thresholds $t_z$ are different for the CV constraint and $p$-% rule.*

*Proof.* Suppose that the optimal solution under the CV or $p$-% rule constraint classifies proportions $q_a$ and $q_b$ of the advantaged and disadvantaged groups, respectively, to the positive class. As shown in Corbett-Davies et al. [10], we can rewrite the immediate utility as

$$u(d, 0.5) = \mathbb{E}[d(X, Z)(p_{Y|X,Z} - 0.5)].$$

Thus the utility will be maximized when $d^*(X, Z) = 1$ for the $q_z$ proportion of individuals in each group that have the highest values of $p_{Y|X,Z}$. Since the optimal values of $q_z$ may differ between the CV-constrained solution and the $p$-% solution, the optimal thresholds may differ as well. $\qquad\square$

Our final result shows that a decision rule that does not directly use $z$ as an input variable or for determining thresholds will have lower accuracy than the optimal rule that uses this information. That is, we show that DLPs are suboptimal for trading off accuracy and impact parity.

**Theorem 4.** *Let $d^*(\mathbf{x}, z)$ be the optimal decision rule under a the CV-$\gamma$ or $p$-% constraint. Let $d_{DLP}(\mathbf{x})$ be the optimal solution to a DLP. If $d(\mathbf{x}, z)$ and $d_{DLP}(\mathbf{x})$ satisfy CV or $p$-% constraints with the same $q_a$ and $q_b$, the DLP solution results in lower or equal accuracy (equal only if the solutions are the same.)*

*Proof.* From Proposition 3, we know that the unique accuracy-optimizing solution is given by $d^*(\mathbf{x}, z) = \delta(p_{Y|X,Z}(\mathbf{x}, z) \geq t_z)$, where $t_z$ is the 1 - $q_z$ quantile of $p_{Y|X,Z}$. The difference in immediate utility between the two decision rules can be expressed as follows:

$$\mathbb{E}[d^*(X, Z)(p_{Y|X,Z} - .5)] - \mathbb{E}[d_{DLP}(X)(p_{Y|X,Z} - .5)]$$
$$= \mathbb{E}[(d^*(X, Z) - d_{DLP}(X))(p_{Y|X,Z} - 0.5)]$$
$$= \mathbb{E}[p_{Y|X,Z} - .5 | d^* = 1, d_{DLP} = 0]\mathbb{P}(d^* = 1, d_{DLP} = 0) - \mathbb{E}[p_{Y|X,Z} - .5 | d^* = 0, d_{DLP} = 1]\mathbb{P}(d^* = 0, d_{DLP} = 1)$$
$$= \left(\mathbb{E}[p_{Y|X,Z} - .5 \mid d^* = 1, d_{DLP} = 0] - \mathbb{E}[p_{Y|X,Z} - .5 \mid d^* = 0, d_{DLP} = 1]\right)\mathbb{P}(d^* = 1, d_{DLP} = 0)$$
$$\geq 0$$

The final inequality follows since $d^*(X, Z) = 1$ for the highest values of $p_{Y|X,Z}$, so $p_{Y|X,Z}$ is stochastically greater on the event $\{d^* = 1, d_{DLP} = 0\}$ than on $\{d^* = 0, d_{DLP} = 1\}$. Note that equality holds only if $\mathbb{P}(d^* = 1, d_{DLP} = 0) = 0$, i.e., if the two rules are (almost surely) equivalent. $\square$

Our results continue to hold under "do no harm" constraints, where we require that any individual in the disadvantaged group who was classified as positive under the unconstrained rule $d_{\text{uncons}}(\mathbf{x}, z)$ remains positively classified. This corresponds to the setting where the proportion of cases in the disadvantaged group classified as positive is constrained to be no lower than the proportion under the unconstrained rule (or no lower than some fixed value $q_a^{\min}$). Such constraints impose an upper bound on the optimal thresholds $t_b$, but do not change the structure of the optimal rules.

**Functional equivalence when protected characteristic is redundantly encoded.** Consider the case where the protected feature $z$ is redundantly encoded in the other features $\mathbf{x}$. More precisely, suppose that there exists a known subcomputation $g$ such that $z = g(\mathbf{x})$. This allows for any function of the data $f(\mathbf{x}, z)$ to be represented as a function of $\mathbf{x}$ alone via $\tilde{f}(\mathbf{x}) = f(\mathbf{x}, g(\mathbf{x}))$. While it remains the case that $\tilde{f}(\mathbf{x})$ does not directly use $z$ as an input variable—and thus satisfies treatment parity—$\tilde{f}$ should be no less legally suspect from a *disparate treatment* perspective than the original function $f$ that uses $z$ directly. The main difference for the purpose of our discussion is that $\tilde{f}$, resulting from a DLP, may technically satisfy treatment parity, while $f$ does not.

While this form of "strict" redundancy is unlikely, characterizing this edge case is important for considering whether DLPs should have different legal standing vis-a-vis disparate treatment than methods that use $z$ directly. This is particularly relevant if one thinks of the 'practitioner' in question as having discriminatory intent. Furthermore, the partial encoding of the protected attribute is commonplace in settings where discrimination is a concern (as with gender in our experiment in §4). Indeed, the very premise of DLPs requires that $\mathbf{x}$ is significantly correlated with $z$. Moreover, DLPs provide an incentive for practitioners to game the system by adding features that are predictive of the protected attribute but not necessarily of the outcome, as these would improve the DLP's performance.

**Within-class discrimination when protected characteristic is partially redundantly encoded.** When the protected characteristic is partially encoded in the other features, disparate treatment may induce within-class discrimination by applying the benefit of the affirmative action unevenly, and can even harm some members of the protected class. Next we demonstrate this phenomenon empirically using (synthetically biased) university admissions data and several public datasets. The ease of producing such examples might convince the reader that the varied effects of intervention with a DLP on members of the disadvantaged group raises practice and policy concerns about DLPs.

## 4 Empirical Analysis

This preceding analysis demonstrates several theoretical advantages to increasing impact parity via treatment disparity:

- **Optimality:** As demonstrated for CV score and for $p$-% rule, intervention via per-group thresholds maximizes accuracy subject to an impact parity constraint.
- **Rational ordering:** Within each group, individuals with higher probability of belonging to the positive class are always assigned to the positive class ahead of those with lower probabilities.
- **Does no harm to the protected group:** The treatment disparity intervention can be constrained to only benefit members of the disadvantaged class.

DLPs attempt to produce a classifier that satisfies the parity constraints, by relying upon the proxy features to satisfy the parity metric. Typically, this is accomplished either by introducing constraints to a convex optimization problem, or by adding a regularization term and tuning the corresponding hyper-parameter. Because the CV score and $p$-% rule are non-convex in model parameters (scores only change when a point crosses the decision boundary), [4, 5] introduce convex surrogates aimed at reducing the correlation between the sensitive feature and the prediction.

These approaches presume that the proxy variables contain information about the sensitive attribute. Otherwise, the parity could only be satisfied via a trivial solution (e.g. assign either *everyone* or *nobody* to the positive class). So we must consider two scenarios: (i) the proxy variables $\mathbf{x}$ fully encode $z$, in which case, a sufficiently powerful DLP will implicitly reconstruct $z$, because this gives the optimal solution to the impact-constrained objective; and (ii) $\mathbf{x}$ doesn't fully capture $z$, or the DLP is unable to recover $z$ from $\mathbf{x}$, in which case the DLP may be sub-optimal, may violate rational ordering within groups, and may harm members of the disadvantaged group.

### 4.1 Synthetic data example: work experience and hair length in hiring

To begin, we illustrate our arguments empirically with a simple synthetic data experiment. To construct the data, we sample $n_{\text{all}} = 2000$ total observations from the data-generating process described below. 70% of the observations are used for training, and the remaining 30% are reserved for model testing.

$$z_i \sim \text{Bernoulli}(0.5)$$
$$\text{hair\_length}_i \mid z_i = 1 \sim 35 \cdot \text{Beta}(2, 2)$$
$$\text{hair\_length}_i \mid z_i = 0 \sim 35 \cdot \text{Beta}(2, 7)$$
$$\text{work\_exp}_i \mid z_i \sim \text{Poisson}(25 + 6z_i) - \text{Normal}(20, \sigma = 0.2)$$
$$y_i \mid \text{work\_exp} \sim 2 \cdot \text{Bernoulli}(p_i) - 1,$$
$$\text{where } p_i = 1/\left(1 + \exp[-(-25.5 + 2.5\text{work\_exp})]\right)$$

This data-generating process has the following key properties: (i) the historical hiring process was based solely on the number of years of work experience; (ii) because women on average have fewer years of work experience than men (5 years vs. 11), men have been hired at a much higher rate than women; and (iii) women have longer hair than men, a fact that was irrelevant to historical hiring practice.

Figure 1 shows the test set results of applying a DLP to the available historical data to equalize hiring rates between men and women. We apply the DLP proposed by Zafar et al. [5], using code available from the authors.[2] While the DLP nearly equalizes hiring rates (satisfying a 105-% rule), it does so through a problematic within-class discrimination mechanism. The DLP rule advantages individuals with longer hair over those with shorter hair and considerably longer work experience. We find that several women who would have been hired under historical practices, owing to their 12+ years of work experience, would not be hired under the DLP due to their short hair (i.e., their male-like characteristics captured in $\mathbf{x}$). Similarly, several men, who would not have been hired based on work experience alone, are advantaged by the DLP due to their longer hair (i.e., their 'female-like' characteristics in $\mathbf{x}$). The DLP violates rational ordering, and harms some of the most qualified individuals in the protected group. Group parity is achieved at the cost of individual unfairness.

Granted, we might not expect factors such as hair length to knowingly be used as inputs to a typical hiring algorithm. We construct this toy example to illustrate a more general point: since DLPs do not have direct access to the protected feature, they must infer from the other features which people are most likely to belong to each subgroup. Using the protected feature directly can yield more

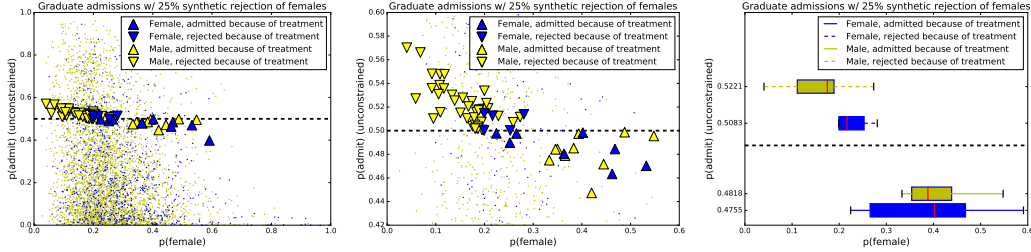

Figure 2: (left) probability of the sensitive variable versus (unconstrained) admission probability, on unseen test data. Downward triangles indicate individuals rejected *only* after applying the DLP ("treatment"), while upward triangles indicate individuals accepted *only* by the DLP. The remaining ∼4,000 blue/yellow dots indicate people whose decisions are not altered. Many students benefiting from the DLP are males who 'look like' females based on other features, whereas females who 'look like' males are hurt by the DLP. Detail view (center) and summary statistics (right) of the same plot.

reasonable policies: For example, by applying per-group thresholds, we could hire the highest rated individuals in each group, rather than distorting rankings within groups based on how female/male individuals *appear* to be from their other features.

## 4.2   Case study: Gender bias in CS graduate admissions

For our next example, we demonstrate a similar result but this time by analyzing real data with synthetic discrimination, to empirically demonstrate our arguments. We consider a sample of ∼9,000 students considered for admission to the MS program of a large US university over an 11-year period. Half of the examples are withheld for testing. Available attributes include basic information, such as country of origin, interest areas, and gender, as well as quantitative fields such as GRE scores. Our data also includes a label in the form of an 'above-the-bar' decision provided by faculty reviewers. Admission rates for male and female applicants were observed to be within 1% of each other. So, to demonstrate the effects of DLPs, we corrupt the data with *synthetic discrimination*. Of all women who were admitted, i.e., $z_i = b, y_i = 1$, we flip 25% of those labels to 0: giving noisy labels $\bar{y}_i = y_i \cdot \eta$, for $\eta \sim Bernoulli(.25)$. This simulates historical bias in the training data.

We then train three logistic regressors: (1) To predict the (synthetically corrupted) labels $\bar{y}_i$ from the non-sensitive features $x_i$; (2) The same model, applying the DLP of [5]; and (3) A model to predict the sensitive feature $z_i$ from the non-sensitive features $x_i$. The data contains limited information that predicts gender, though such predictions can be made better than random (AUC=0.59) due to different rates of gender imbalance across (e.g.) countries and interests.

Figure 2 (left) shapes our basic intuition for what is happening: Considering the probability of admission for the unconstrained classifier (y-axis), students whose decisions are 'flipped' (after applying the fairness constraint) tend to be those close to the decision boundary. Furthermore, students *predicted* to be male (x-axis) tend to be flipped to the negative class (left half of plot) while students *predicted* to be female tend to be flipped to the positive class (right half of plot). This is shown in detail in Figure 2 (center and right). Of the 43 students whose decisions are flipped to 'non-admit,' 5 are female, each of whom has 'male-like' characteristics according to their other features as demonstrated in our synthetic hair-length example. Demonstrated here with real-world data, the DLP both disrupts the within-group ordering and violates the *do no harm* principle by disadvantaging some women who, but for the DLP, would have been admitted.

**Comparison with Treatment Disparity.**   To demonstrate the better performance of per-group thresholding, we implement a simple decision scheme and compare its performance to the DLP.

Our thresholding rule for maximizing accuracy subject to a $p$-% rule works as follows: Recall that the $p$-% rule requires that $q_b/q_a > p/100$, which can be written as $\frac{p}{100}q_a - q_b < 0$. We denote the quantity $\frac{p}{100}q_a - q_b$ as the $p$-gap. To maximize accuracy subject to satisfying the $p$-% rule, we construct a score that quantifies reduction in $p$-gap per reduction in accuracy. Starting from the

Table 1: Statistics of public datasets.

| dataset | source | protected feature | prediction target | $n$ |
|---|---|---|---|---|
| Income | UCI [22] | Gender (female) | income > \$50k | 32,561 |
| Marketing | UCI [23] | Status (married) | customer subscribes | 45,211 |
| Credit | UCI [24] | Gender (female) | credit card default | 30,000 |
| Employee Attr. | IBM [25] | Status (married) | employee attrition | 1,470 |
| Customer Attr. | IBM [25] | Status (married) | customer attrition | 7,043 |

accuracy-maximizing classifications $\hat{y}$ (thresholding at .5), we then flip those predictions which close the gap fastest:

1. Assign each example with $\{\tilde{y}_i = 0, z_i = b\}$ or $\{\tilde{y}_i = 1, z_i = a\}$, a score $c_i$ equal to the reduction in the p-gap divided by the reduction in accuracy:
   (a) For each example in group $a$ with initial $\hat{y}_i = 1$, $c_i = \frac{p}{100 n_a (2\hat{p}_i - 1)}$.
   (b) For each example in group $b$ with initial $\hat{y}_i = 0$, $c_i = \frac{1}{n_b (1 - 2\hat{p}_i)}$.
2. Flip examples in descending order according to this score until the desired CV-score is reached.

These scores do not change after each iteration, so the greedy policy leads to optimal flips (equivalently, optimal classification thresholds).

The unconstrained classifier achieves a p-% rule of 71.4%. By applying this thresholding strategy, we were able to obtain the same accuracy as the method of [5], but with a higher $p$-% rule of 78.3% compared to 77.6%. Note that on this data, the method of [5] maxes out at a $p$-% rule of 77.6%. That is, the method is limited in what $p$-% rules may be achieved. By contrast, the thresholding rule can achieve any desired parity level. Subject to a $< 1\%$ drop in accuracy relative to the DLP we can achieve a $p$-% rule of $\sim 100\%$.

## 4.3 Examples on public datasets

Finally, for reproducibility, we repeat our experiments from Section 4.2 on a variety of public datasets (code and data will be released at publication time). Again we compare applying our simple thresholding scheme against the fairness constraint of [5], considering a binary outcome and a single protected feature. Basic info about these datasets (including the prediction target and protected feature) is shown in Table 1.

The protocol we follow is the same as in Section 4.2. Each of these datasets exhibits a certain degree of bias w.r.t. the protected characteristic (Table 2), so no synthetic discrimination is applied. In Table 2, we compare (1) The $p$-% rule obtained using the classifier of [5] compared to that of a naïve classifier (column k vs. column h); and (2) The $p$-% rule obtained when applying our thresholding strategy from Section 4.2. As before, half of the data are withheld for testing.

First, we note that in most cases, the method of [5] increases the $p$-% rule (column k vs. h), while maintaining an accuracy similar to that of unconstrained classification (column i vs. f). One exception is the UCI-Credit dataset, in which *both* the accuracy and the $p$-% rule simultaneously decrease; although this is against our expectations, note that the optimization technique of [5] is an approximation scheme and does not offer accuracy guarantees in practice (nor can it in general achieve a $p$-% rule of 100%). However these details are implementation-specific and not the focus of this paper. Second, as in Section 4.2, we note that the optimal thresholding strategy is able to offer a strictly larger $p$-% rule (column l vs. k) at a given accuracy (in this case, the accuracy from column i). In most cases, we can obtain a $p$-% rule of (close to) 100% at the given accuracy.

We emphasize that the goal of our experiments is not to 'beat' the method of [5], or even to comment on any specific discrimination-aware classification scheme. Rather, we emphasize that *any* DLP is fundamentally upper-bounded (in terms of the $p$-% rule/accuracy trade-off) by simple schemes that explicitly consider the protected feature. Our experiments validate this claim, and reveal that the two schemes make strikingly different decisions. While concealing the protected feature from the classifier may be conceptually desirable, practitioners should be aware of the consequences.

Table 2: Comparison between unconstrained classification, DLPs, and thresholding schemes. Note that the $p$-% rules from [5] were the strongest that could be obtained with their method; on complex datasets $p$-% rules of 100% are rarely obtained in practice, due to their specific approximation scheme. Employee and Customer datasets are from IBM, the others are UCI datasets.

| basic statistics | | | | | naïve (unconstrained) classification | | | fair (constrained) classification [5] | | | optimal threshold |
|---|---|---|---|---|---|---|---|---|---|---|---|
| dataset | %prot. | %prot. in +'ve | %non-prot. in +'ve | label $p$-% | acc. | prot./non-prot. in positive | $p$-% | acc. | prot./non-prot. in positive | $p$-% | $p$-% at const. acc. |
| a | b | c | d | e | f | g | h | i | j | k | l |
| Income | 66.9% | 30.6% | 10.9% | 35.8% | 0.85 | 8% / 25% | 31% | 0.85 | 7% / 24% | 29% | 52.9% |
| Marketing | 60.2% | 14.1% | 10.1% | 71.9% | 0.89 | 3% / 4% | 82% | 0.89 | 3% / 3% | 102% | 100.3% |
| Credit | 60.4% | 24.1% | 20.8% | 86.0% | 0.82 | 10% / 12% | 88% | 0.74 | 21% / 25% | 85% | 100.0% |
| Employee | 45.8% | 19.2% | 12.5% | 65.0% | 0.87 | 8% / 12% | 65% | 0.86 | 8% / 11% | 69% | 100.4% |
| Customer | 48.3% | 33.0% | 19.7% | 59.7% | 0.80 | 15% / 30% | 49% | 0.79 | 16% / 19% | 84% | 100.2% |

# 5    Discussion

**Coming to terms with treatment disparity.**    Legal considerations aside, treatment disparity approaches have three advantages over DLPs: they optimally trade accuracy for representativeness, preserve rankings among members of each group, and do no harm to members of the disadvantaged group. In addition, treatment disparity has another advantage: by setting class-dependent thresholds, it's easier to understand how treatment disparity impacts individuals. It seems plausible that policy-makers could reason about thresholds to decide on the right trade-off between group equality and individual fairness. By contrast the tuning parameters of DLPs may be harder to reason about from a policy standpoint. Several key challenges remain. Our theoretical arguments demonstrate that thresholding approaches are optimal in the setting where we assume complete knowledge of the data-generating distribution. It is not always clear how best to realize these gains in practice, where imbalanced or unrepresentative datasets can pose a significant obstacle to accurate estimation.

**Separating estimation from decision-making.**    In the context of algorithmic, or algorithm-supported decision-making, it's often useful to obtain not just a classification, but also an accurate probability estimate. These estimates could then be incorporated into the decision-theoretic part of the pipeline where appropriate measures could be taken to align decisions with social values. By intervening at the modeling phase, DLPs distort the predicted probabilities themselves. It's not clear what the outputs of the resulting classifiers actually signify. In unconstrained learning approaches, even if the label itself may reflect historical prejudice, one at least knows what is being estimated. This leaves open the possibility of intervening at decision time to promote more equal outcomes.

**Fairness beyond disparate impact**    How best to quantify discrimination and unfairness remains an important open question. The CV scores and $p - \%$ rules offer one set of definitions, but there are many other parity criteria to which our results do not directly apply, e.g., equality of opportunity [13]. Other notions of fairness and the trade-offs between them have been studied [14, 26–29]. In a recent paper, Zafar et al. [30] depart from parity-based definitions and propose instead a preference-based notion of fairness. Dwork et al. [11] address the problem of how best to incorporate information about protected characteristics for several of these other fairness criteria.

Problematically, research into fairness in ML is often motivated by the case in which our ground-truth data is tainted, capturing existing discriminatory patterns. Characterizing different forms of data bias, how to detect them, and how to draw valid inference from such data remain important outstanding challenges.

Even in settings where treatment disparity in favor of disadvantaged groups is an acceptable solution, questions remain of "how", "how much?" and "when?". While in some cases treatment disparity may arguably be correcting for omitted variable bias historical discrimination, in other settings it may be viewed as itself a form of discrimination. For example, in the United States, Asian students are simultaneously over-represented and discriminated against in higher education [2]. Such policy judgments require a keen understanding and awareness of the social and historical context in which the algorithms are developed and meant to operate. Recent work on identifying proxy discrimination [31] and causal formulations of fairness [32–34] offer some promising approaches to translating such understanding into technological solutions.

## Footnotes

[1]Our results are all presented in terms of a more general performance metric, of which classification accuracy is a special case.

[2]https://github.com/mbilalzafar/fair-classification/

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
