[Reviews · NeurIPS 2018]

Reviewer 1



This paper tackles a class of algorithms defined as Disparate Learning Processes (DLP) which use the sensitive feature while training and then make predictions without access at the sensitive feature. DLPs have appeared in multiple prior works, and the authors argue that DLPs do not necessarily guarantee treatment parity, which could then hurt impact parity. The theoretical analysis focuses on relating treatment disparity to utility and then optimal decision rules for various conditions. Most notably the per-group thresholding yields optimal rules to reduce the CV gap. As outlined in the beginning of section 4, the theoretical advantages of DLPs seems to optimality, rational ordering, and "no additional harm" to the protected group. In the experimental section, the authors use a corrupted version of one dataset (CS grad admissions) and five public datasets (many of whom are drawn from the UCI dataset collection). For the grad admission dataset, DLP violate within-group ordering and "no additional harm" criteria. Comparing the DLP with a thresholding decision scheme, the authors show that DLPs may cause both accuracy and impact disparity (CV gap) to decrease. I enjoyed this paper and its discussion of treatment parity and impact parity. In particular, the legal background helps ground the competing definitions of parity which therefore extend outward into classes of classifiers. At the same time, parsing out the implied definitions was surprisingly difficult. The paper would be strengthened by formal mathematical definitions of treatment parity and impact parity early in the paper. Without rigorous definitions, it was difficult to align the legal definitions in the introductions and the claims later in the paper. For example, are the criteria listed in the beginning of section 4 included in the definitions of either treatment parity or impact parity -- or simply other fairness criteria that the authors chose to consider? A more minor thought, but the title of the paper asks whether impact disparity requires treatment disparity. This question is not answered in the paper since the paper only addresses DLPs, specifically how DLPs claim to address both impact and treatment disparity but may in fact fail. As I understand it, if there is another method out there that satisfies impact disparity without treatment disparity, we have not yet disproven its existence.

Reviewer 2



This paper provides both a theoretical and empirical analysis of disparate learning processes (DLPs). Disparate learning processes seek to learn "fair" decision rules by taking into account sensitive attributes at training time but producing rules that do not depend on sensitive attributes. Building on existing results, the authors show that for two natural measures of disparity between group outcomes, the optimal way to maximize accuracy subject to a disparity threshold is to set group-specific threshold. They go on to show experimentally the cost of using DLPs, which cannot use information about the sensitive attribute at decision time, compared to group-specific thresholds. As expected, the group-specific thresholds outperform DLPs. Some experiments are done on data that isn't publicly available, though the authors do show similar results on publicly available datasets. The paper is well-written and easy to understand. I find the results in this paper to be underwhelming. The theoretical results follow almost directly from Corbett-Davies et al. (reference [10] in the paper), and they imply that DLPs must be suboptimal, which the experiments go on to show. There is an argument to be made for the value of the experiments demonstrating the degree to which DLPs are suboptimal, and this to me is main contribution of the paper; however, this in itself doesn't seem to be particularly compelling. One interesting direction that the paper touches on is the legal argument for using DLPs. Given that they're clearly suboptimal at reducing the particular forms of disparity being considered here, it's natural to ask why one would consider them at all. Here, the authors claim that DLPs are somehow aligned with the ruling in Ricci v. DeStefano, which set precedent for designing and administering decision-making rules in hiring contexts. This seems to be crucial point: if DLPs are legal in some contexts in which group-specific thresholds are not, then they could be in some sense Pareto-optimal in those contexts; however, if there are no such contexts, then it doesn't seem particularly relevant to investigate how suboptimal they are. This may be outside the scope of this paper, but it would be interesting to see at least some discussion or justification for the use of DLPs. The biggest factor in my review is the lack of novelty in this work -- in my view, most of the theoretical results are easily derived from previous work, and the empirical results are fairly unsurprising.

Reviewer 3



Summary: This paper describes a series of negative results for "disparate learning processes" (DLPs). These are recent methods to train classification models whose predictions obey a given fairness constraint between protected subgroups (e.g., by solving a convex optimization problem or preprocessing). The key characteristic of DLPs is that they have access to labels for the protected groups at training time, but not at deployment. In practice, this means that a DLP will produce a single model for the entire population. In this paper, the authors present several arguments against DLPs, including: 1. If the protected attribute is redundantly encoded in other features, then a DLP may result in the same treatment disparity that aims to overcome. 2. If the protected attributed is partially encoded in other features, then a DLP may induce within class-discrimination and harm certain members of the protected group. 3. A DLP provides suboptimal accuracy in comparison to other classifiers that satisfy a specific family of fairness constraints. The authors present several proofs of the theorems for an idealized setting. They complement the theorems with a series of empirical results on a real-world dataset and UCI datasets showing the shortcomings of DLPs and the superiority of simple alternatives (e.g. training a logistic regression model and applying per-group thresholding). Comments: I thought that this is a well-written paper on an interesting and timely topic. I think that it is important to state that the central contribution of this paper is NOT methodological novelty or theoretical novelty, but rather the results against DLPs. I believe that these results will be significant in the long-term as much recent work in this area has focused on tackling fairness-related issues through DLPs without considering their shortcomings. I believe that the current paper would significantly strengthened if the authors were to consider broader approaches to treatment disparity. For example, many readers would consider Dwork et al's proposed approach to train “separate models for separate groups” as a natural extension to the per-group thresholding used in the numerical experiments. The authors currently cite these methods in the text under the “alternative approaches” in Section 2 and the “fairness beyond disparate impact” in Section 5. However, they should be discussed more thoroughly and included in the experimental results in Section 4. Other than this, two other major points to address: - Among the arguments presented against DLPs, 1. deserves some qualification. It is unlikely that a practitioner would not check for that the protected attribute is redundantly encoded before training the model. - The results in Section 3 apply to a setting where the authors have access to the optimal classifier d*. This is fine given that the results are in line with their empirical work in Section 4. However, the authors should explicitly state that this does not capture the real-world setting (i.e., where the classifier is learned from data). There should also be a discussion as to when can should expect the results to manifest themselves in the real-world setting Specific Comments: - The main findings (e.g. points 1. through 4) at the end of the Introduction are great. I would recommend adding a short example after each point so that readers from a broader audience will be able to understand the points without having to read the papers. - Proofs in Section 3 could be put into an Appendix for the sake of space. - The “Complicating matters…” (line 29) in the Introduction is interesting and merits further discussion. I’d suggest placing some of the discussion from lines 318 – 326 here. - “Alternative Approach” in Section 2 and the “Fairness beyond Disparate Impact” in Section 5 should be merged (see also comments above) - Section 2: 0-1 loss should be defined in line 70. I'd recommend changing the Section title to Preliminaries. Minor Comments: - Table 2: the formatting can be improved. The column labels should be in a different - Line 6 “\textit{dis}parity” <- This seems like a typo at first. I'd recommend removing it since it just adds a play on words. - Line 103 "in the next section (\S 4)" <- In Section 4 - Notation: for clarity, I’d recommend using $h$ for the model and \hat{y} for the predicted value. - Line 298 "data sets" <- "datasets" - Line 299 “separating estimation and decision-making” <- “separating estimation from decision-making” - Line 310 “In a recent” <- “In recent” Post Rebuttal ========== I have raised my score as the authors have agreed to address some of the issues raised in my original review. The current submission does not fit the mold of a traditional NIPS paper. However, it provides a crisp and thoughtful discussion on the technical limitations of DLPs. I believe that this makes for an important contribution, especially given that DLPs have become a standard approach to handle issues related to fairness in ML (see e.g. Aggarwal et al. and Kilbertus et al. from ICML 2018). In light of this, I’d encourage the authors to make their work accessible to as broad an audience as possible. Aside from issues raised in my original review, I think one other issue that could be addressed (and brought up by other reviewers) is to answer the question they ask in the title in more direct manner (or, alternatively, to change their title).